# Changes in the Compressive Strength of Concrete in Thin Horizontally Formed Slabs

**DOI:** 10.3390/ma13245671

**Published:** 2020-12-11

**Authors:** Jacek Szpetulski, Bohdan Stawiski

**Affiliations:** 1Faculty of Civil Engineering, Mechanics and Petrochemistry, Warsaw University of Technology, 09-400 Płock, Poland; 2Faculty of Environmental Engineering and Geodesy, Wroclaw University of Environmental and Life Sciences, 50-363 Wrocław, Poland; bohdan.stawiski@upwr.edu.pl

**Keywords:** destructive test, concrete compressive strength, slab, core sample, concrete mix

## Abstract

During compaction of a concrete mix, when thin slabs are formed in a horizontal position, the components of this mix become segregated. Heavy components fall to the bottom, and light components (air and water) move to the top. This process may suggest that the upper layers of concrete elements formed in a horizontal position may have lower compressive strength than the remaining part of the element. This problem is recognized and documented in many publications, but there was a publication whose test results indicate a lack of variability in the compressive strength of concrete across the thickness of tested elements. The discrepancies appearing in the evaluation of concrete homogeneity was the reason for conducting destructive tests of the compressive strength of concrete across the thickness of horizontally concreted test elements that imitate thin slabs. The obtained results of the destructive compressive strength confirmed previous results regarding the heterogeneity of concrete. They clearly indicate that there is a differentiation of the compressive strength of concrete across the thickness of a thin element, which remained in a liquefied state for a certain time during its formation. The longer the duration of this state across the entire thickness of the formed element, the greater the differentiation of the compressive strength between the top and bottom layers.

## 1. Introduction

Thin horizontally formed concrete slabs include industrial floors, the thickness of which does not exceed 30 cm, with 20 cm often being sufficient. Monolithic ceiling slabs installed in buildings, especially residential ones, are thin and have a thickness of up to 20 cm. It is necessary to distinguish thin slabs from those that are relatively thick, e.g., ones with beams and binding joists, the thickness of which usually exceeds 30 cm. Thick elements and columns are formed in layers, whereas the execution of floors and monolithic ceiling slabs depends on the decision of the contractor. This is important due to the segregation of components in the concrete mix, which takes place during the vibration that causes its liquefaction. Heavy components fall to the bottom, and light components (air and water) move to the top. In elements formed in layers, the weakest top zone in the layer, which is already compacted, mixes with the strongest bottom zone of the subsequent layer. This causes the differentiation of the compressive strength of the concrete to be reduced to some extent, i.e., homogenization of the concrete. The described process appeared in the research presented in the publication [1]. The research was conducted on small test elements with the recommendation to fabricate them as homogeneously as possible. The tests about the heterogeneity of concrete were conducted on test elements made in laboratory conditions from which many cores were taken in the vertical and horizontal directions. The task was completed with a satisfactory effect [1], which was confirmed by the results of the compressive strength of concrete. The test results presented in publication [1] were meant to confirm the old and outdated knowledge concerning the homogeneity of concrete.

The reasons presented above became the basis for the further analysis of the generally available literature regarding this topic, as well as for the undertaking of research that aims to explain the apparent contradiction. Compressive strength in horizontally formed concrete increases towards the bottom layer of the element [2]. Similar information can also be found in the information in Appendix C of standard [3]: “With height of a concrete pour, in-situ strength decreases toward the top of a pour, even for slabs, and can be up to 25% less at the top than in the body of the concrete.“

In studies [4,5,6,7,8,9,10,11,12,13,14,15,16], it was proven that the differentiation of the compressive strength of concrete occurs across the thickness of thin horizontally formed slabs. The opinions of various researchers concern the size of the difference in the compressive strength between the various levels at which it is tested. The biggest differences exist between the extreme layers of the bottom and top of a slab. Tests regarding the compressive strength of concrete in the most extreme layers are only possible when using indirect methods, e.g., the ultrasonic method. An attempt can be made to answer the question regarding the differentiation of the compressive strength of concrete in the extreme top and bottom layers of thin slabs by testing smaller and smaller core samples. The compressive strength of a given core sample is equal to the compressive strength in the middle of its height [10], and therefore, in order to determine the compressive strength of core samples with a diameter of 100 mm, the tests need to be performed at a distance of no less than 50 mm. If the core sample has a diameter of 50 mm and is cut from the lowest layer of the slab, the compressive strength can be determined at a distance of 25 mm from the bottom or top of the slab using the destructive method. Unfortunately, the reduction of the core sample size is not unlimited because its size is connected to the size of the coarse aggregate, as described in standard [17]: “The aggregate size has a significant influence on the measured strength when the core diameter divided by the upper aggregate size is less than about 3.“ When using small diameter drills, the increased frictional force between the concrete sample and the pressure plates of the testing machine, which stops the sample being destroyed, should also be considered [18,19,20,21].

The considered value of compressive strength and its variability across the structure’s height can be influenced by various factors, including the method of compaction, vibration time, consistency of the concrete mix, aggregate type, etc. Therefore, the testing of compressive strength on core samples for cores with various diameters was planned, and the sample elements for the tests were made with different compositions and by using different compaction technologies.

## 2. Materials and Methods 

In order to verify the variation of the compressive strength of concrete tested using the destructive method across the thickness of a horizontally formed sample element, a research project was undertaken in which the following was planned:

I. The development of recipes of concrete mixes for concrete within three compressive strength ranges: 20 MPa, 40 MPa and 60 MPa. 

In order to make the concrete mixes for the three concretes with the assumed compressive strengths, the following was used:Cement CEM II/B-V 32,5R;Clean natural aggregate, grain size 0 ÷ 2 mm (river sand);Clean natural aggregate, grain size 0 ÷ 16 mm (gravel from the mine);Potable water taken from the municipal waterlines;Plasticizer FK-88.

The compressive strength test was conducted according to standard [22] on standard cubic samples with side dimensions of 150 mm, which were prepared according to standard [23]. The concrete with the low compressive strength (20 MPa) was designed with a water/cement (W/C) ratio equal to 0.75. The consistency class of the concrete mix was F5 or S4 according to standard [24] determined on the basis of tests using the following methods: the flow table method according to standard [25], and the concrete slump method according to standard [26]. The concrete with the compressive strength of 40 MPa had a W/C ratio of 0.5 and a concrete mix consistency of F2 or S1 [24]. The concrete with the highest compressive strength of 60 MPa had a W/C ratio of 0.35 and a concrete mix consistency of S1 according to standard [24]. The consistency classes were not the same, which may have had a certain effect on the possibility of segregating the concrete mix components during compaction.

II. The execution of sample elements (thin slabs) with a thickness of 260 mm using three compaction methods: Compaction Method A—the vibrating of the concrete mix with a frequency of 50 Hz on a vibration table, the concrete mix being implemented in two layers, and the vibration taking 2 min for each layer with a time interval of 5 min; Compaction Method B—the vibrating of the concrete mix with a frequency of 50 Hz on the vibration table, the concrete mix being implemented in two layers, and the vibration taking 6 min for each layer with a time interval of 5 min; and Compaction Method C—the vibrating of the concrete mix with a frequency of 300 Hz for a period of 5 min using an immersion vibrator with ϕ 38 mm.

III. The testing of the core samples with a height-to-diameter ratio equal to 1, which were cut from the core samples according to the concreting direction (Figure 1): The samples were cut in such a way that half of the height of each core sample was placed as low as possible in the bottom layer and as high as possible in the top layer, as well as in the middle of the slab’s thickness (Figure 2). After cutting, the core samples were stored for 30 days at a temperature of 20 ± 2 °C and moisture of 0 ± 15% until the time of testing their compressive strength (Figure 3).

Since the sample elements from which the core samples were taken were 260 mm high, the compressive strength of the concrete was tested in the following layers measured from the bottom of the element (Figure 2):29.5 mm (half the height of the core sample with a diameter of 59 mm);47 mm (half the height of the core sample with a diameter of 94 mm);130 mm (center of the slab);213 mm (260 mm minus half the height of the core sample with a diameter of 94 mm);230.5 mm (260 mm minus half the height of the core sample with a diameter of 59 mm).

In the executed sample elements, the compressive strength was tested at five different levels, from which five core samples were taken for each diameter. A disadvantage of the chosen method was the fact that the extreme bottom and top layers, each 29.5 mm thick, were not tested, and these are the areas where the biggest dispersion of compressive strength might be expected.

In order to determine the compressive strength of the core samples, steel disks with neoprene pads (Figure 4) according to standard [27] were used. The applied testing methodology allowed the destructive force to be measured during the compressive strength test of the core samples taken from the extreme layers of the concrete elements without the need to level them according to standard [17]. The use of the steel disks with neoprene pads provides the same compressive strength test results as other techniques of leveling the face surfaces of tested core samples [28].

## 3. Results and Discussion

In order to compare the compressive strength of the concrete at different levels, the values of the average compressive strength were calculated for five core samples (with a diameter of 59 mm and 94 mm) taken from each layer of the slab.

Figure 5 shows the differentiation in concrete’s compressive strength across the thickness of sample elements that were made of the weakest concrete (20 MPa) with S4 consistency. The compressive strength was determined on the core samples with diameters of 59 mm and 94 mm, which were compacted using Methods A, B and C.

The biggest differentiation of compressive strength across the thickness of the tested element made of 20 MPa concrete, which was determined for the core samples with a diameter of 94 mm, was registered in the case of using Compaction Method B on the vibration table:Δf_c,__ϕ__94_ = |f_c,top_ − f_c,bottom_| = |17.72 MPa − 29.65 MPa| = 11.93 MPa,(1)
and the smallest differentiation in the case of using Compaction Method C with the use of the immersion vibrator:Δf_c,__ϕ__94_ = |f_c,top_ − f_c,bottom_| = |19.68 MPa − 27.82 MPa| = 8.14 MPa.(2)

The greatest difference in the compressive strength of concrete between the extreme measurement planes is as high as 11.93 MPa. This means that the top part of the element is weaker than the bottom part of the element by 40.2%. The reduction of the dimensions of the core samples causes an increase in the distance between the measurement planes.

The biggest differentiation of compressive strength across the thickness of the tested element made of 20 MPa concrete, which was determined for the core samples with a diameter of 59 mm, were registered in the case of using Compaction Method B on the vibration table:Δf_c,__ϕ__59_ = |f_c,top_ − f_c,bottom_| = |20.85 MPa − 35.44 MPa| = 14.59 MPa—weakening of the top layer is equal to 41.2%,(3)
and the smallest differentiation was obtained in the case of Compaction Method C conducted using the immersion vibrator:Δf_c,__ϕ__59_ = |f_c,top_ − f_c,bottom_| = |22.81 MPa − 33.54 MPa| = 10.73 MPa—weakening of the top layer is equal to 32.0%.(4)

The above-mentioned results clearly indicate that the level of differentiation of compressive strength between the bottom and surface layers of the slab is affected by the method of compacting the concrete mix and also by the vibration time. Moreover, assuming that concrete at the height of the slab cross-section is homogenous is a distortion of reality.

The consistency of the 40 MPa concrete mix was less fluid due to it being of class S1. Figure 6 presents how the compressive strength of concrete changes across the thickness of the sample elements with regards to the compaction method.

Differentiation of compressive strength across the thickness of the tested element made of 40 MPa concrete, which was determined on the core samples with a diameter of 59 mm, is as follows:Compaction Method A:
Δf_c,__ϕ__59_ = |f_c,top_ − f_c,bottom_| = |42.39 MPa − 52.0 MPa| = 9.61 MPa—weakening of the top layer is equal to 18.5%;(5)Compaction Method B:
Δf_c,__ϕ__59_ = |f_c,top_ − f_c,bottom_| = |41.71 MPa − 57.52 MPa| = 15.81 MPa—weakening of the top layer is equal to 27.5%;(6)Compaction Method C:
Δf_c,__ϕ__59_ = |f_c,top_ − f_c,bottom_| = |44.87 MPa − 54.71 MPa| = 9.84 MPa—weakening of the top layer is equal to 18.0%.(7)

It is more difficult to transform less plastic concrete into a fluid state, and therefore the process of concrete component segregation is weaker. The reduction of the compressive strength of concrete in the top layer is then much lower and can amount to between 18.0% and 27.5%.

The concrete with the highest compressive strength of 60 MPa also had S1 consistency but was less fluid. Figure 7 shows the differentiation of concrete compressive strength in the sample elements tested on the core samples with a small diameter.

Differentiation of compressive strength across the thickness of the tested element made of 60 MPa concrete, which was determined on the core samples with a diameter of 59 mm is as follows:Compaction Method A:
Δf_c,__ϕ__59_ = |f_c,top_ − f_c,bottom_| = |59.77 MPa − 72.34 MPa| = 12.57 MPa—weakening of the top layer is equal to 17.4%;(8)Compaction Method B:
Δf_c,__ϕ__59_ = |f_c,top_ − f_c,bottom_| = |57.14 MPa − 78.71 MPa| = 21.57 MPa—weakening of the top layer is equal to 27.4%;(9)Compaction Method C:
Δf_c,__ϕ__59_ = |f_c,top_ − f_c,bottom_| = |63.69 MPa − 76.42 MPa| = 12.73 MPa—weakening of the top layer is equal to 16.7%.(10)

In the case of the concrete with a barely fluid consistency, the differentiation of compressive strength across the thickness of the sample elements was even lower and ranged from 16.7% to 27.4%.

## 4. Conclusions

The figures presented above result from testing the compressive strength of concrete. The tests were conducted on core samples with different diameters, which were taken from thin slabs made of concrete that had different compressive strengths, differently compacted and formed of concrete mixes with different consistencies. The purpose of the completed research project was to verify previous test results [8] indicate a lack of variability in the compressive strength of concrete on the thickness of tested elements. The results in the publication [8] debunked the results of previous studies [4,5,6,7,8,9,10,11,12,13,14,15,16], in which it was proven that the differentiation of the compressive strength of concrete occurs on the thickness of horizontally formed slabs.

The authors’ tests were performed on core samples taken from nine sample elements, which were formed in a way similar to that of executing actual building elements. The concrete’s parameters (consistency, vibration time, etc.), which change during the preparation and compaction of the concrete mix, were intentionally changed.

The results obtained from the destructive compressive strength tests confirm previous results from ultrasonic tests. They clearly and unequivocally indicate that concretes with a high W/C ratio and significant plasticity show considerable differentiation of their compressive strength across the thickness of a thin sample element, during which formation remained in a liquefied state for a certain time. The longer this state lasts across the entire thickness of the formed element, the greater the differentiation of compressive strength that occurs between the top and bottom layers. The tests proved that the greatest weakening of the top layer was as much as 41.2%.

The designer chooses the different safety formats (i.e., global resistance method and probabilistic method) [29] within the global resistance format for the estimate of the global design strength of RC structures. The safety formats estimate the design global resistance capacities capturing any modification in the failure mode of the RC structure. The global resistance method always provides design ultimate loads lower than that evaluated with the probabilistic method because the probabilistic method takes into account the aleatory uncertainty of the material’s properties [29]. For compressed elements that are calculated with the assumption of aleatory uncertainties and have their compressive strength determined on cubes, appropriate corrections that result from the chosen calculation method are taken into consideration. If the fact that the compression zone is weaker by, e.g., 41.2%, is taken into account, the load-bearing capacity of such a layer in the structure will decrease significantly. In monolithic ceiling slabs, the compression zone occurs in their upper parts, and the compressive strength of concrete is designed for this zone. In practice, this may mean that the concrete in the top zone does not meet the designer’s assumptions.

The concretes made of a less fluid concrete mix did not achieve such a high differentiation of compressive strength across their thickness (the weakening of the top layer ranged from 18.0% to 27.5%). In the concretes with a very low W/C ratio (equal to 0.35) and dry consistency, the differentiation of compressive strength between the bottom and top layers only ranged from 16.7% to 27.4%. In order to reduce the differences in the compressive strength across the thickness of a slab, it is necessary to ensure an appropriate consistency and also a W/C ratio that will not liquefy the concrete mix. Thin structural elements made of concrete must be carefully formed by contractors who have experience in pouring and compacting concrete mixes.

It is unfortunate that there are researchers (such as those mentioned at the beginning of the paper) who, on the basis of one experiment (one sample element), try to disqualify test results concerning the homogeneity of concrete. These results can be found in many sources and have been confirmed and verified many times using various methods.

The systematic variability of concrete, unlike the random variability, should be evaluated using other parameters that refer to the differentiation of the compressive strength of concrete across the thickness of a slab or floor, e.g., the compressive strength gradient, as proposed in publication [13].

The confirmed differentiation of the compressive strength across the thickness of thin concrete slabs should be taken into account when determining the compressive strength of concrete in a structure. Compressive strength should be determined on the basis of core samples taken from different layers of concrete slabs.

## Figures and Tables

**Figure 1 materials-13-05671-f001:**
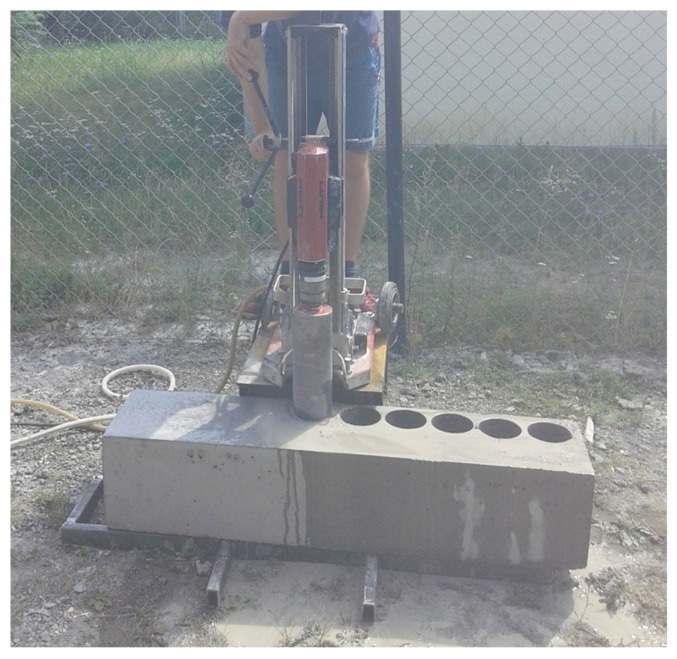
The taking of core samples using a drilling rig with a crown drill on a properly prepared stand.

**Figure 2 materials-13-05671-f002:**
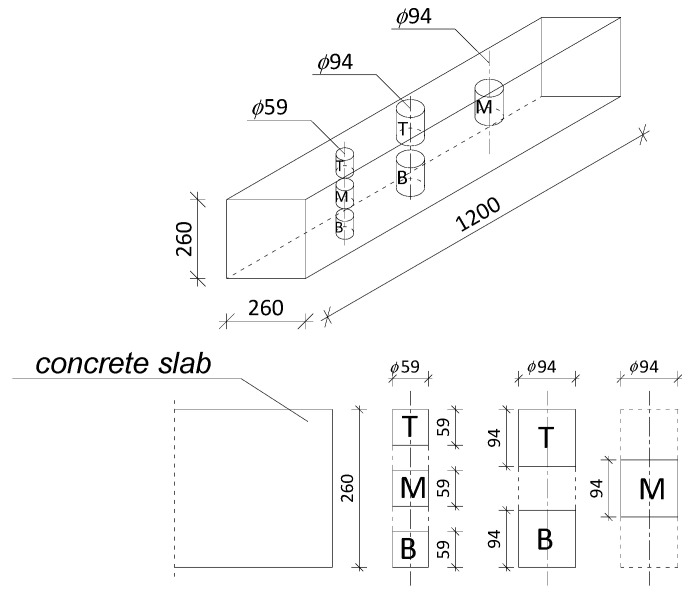
Slab with layers in which the compressive strength of the concrete was tested. T—top layer, M—middle layer, B—bottom layer.

**Figure 3 materials-13-05671-f003:**
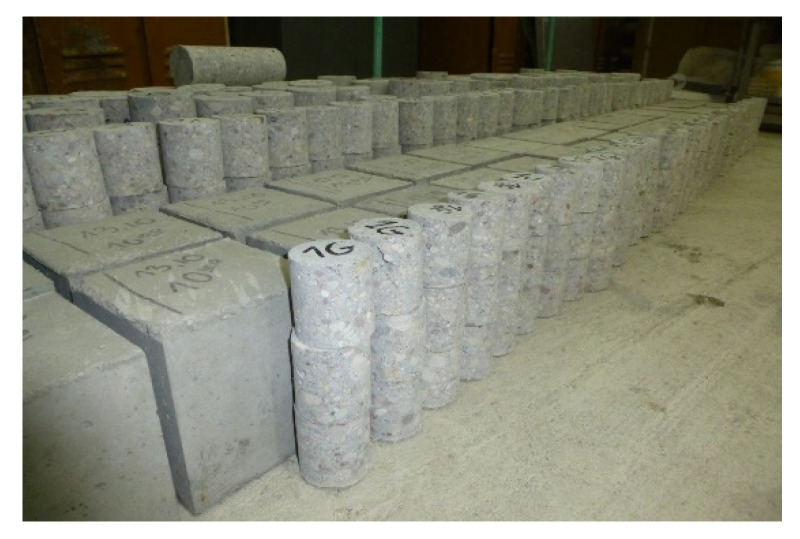
Storing of the core and cubic sample.

**Figure 4 materials-13-05671-f004:**
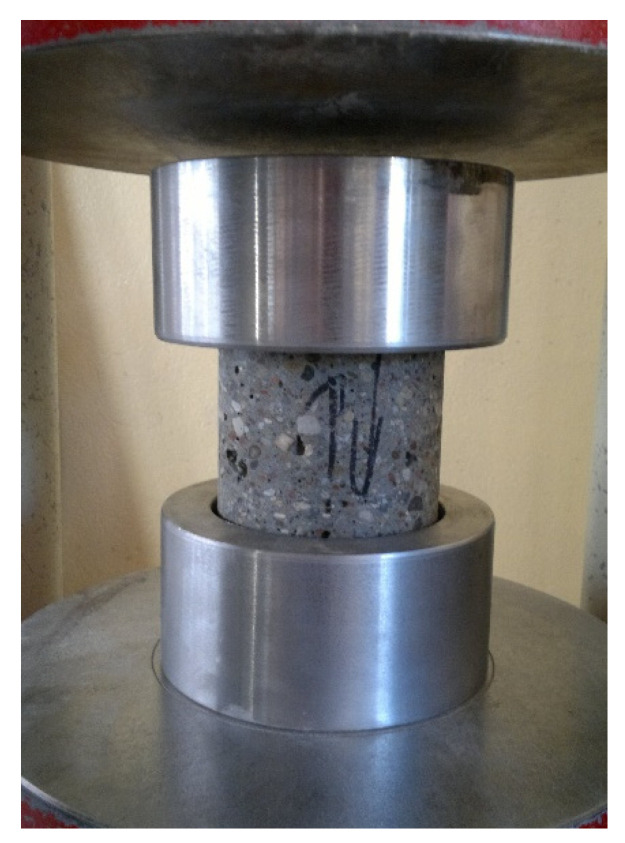
Core sample during the compressive strength test in the strength testing machine.

**Figure 5 materials-13-05671-f005:**
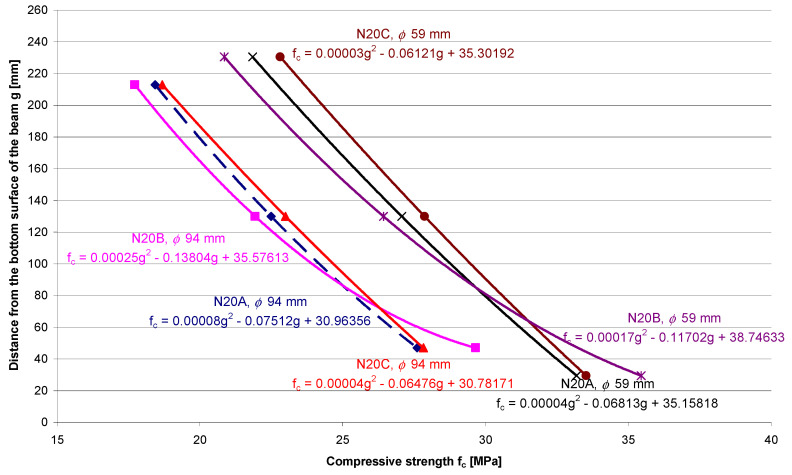
Distribution of the compressive strength of the 20 MPa concrete, which was determined across the height of the core samples with diameters of 59 mm and 94 mm.

**Figure 6 materials-13-05671-f006:**
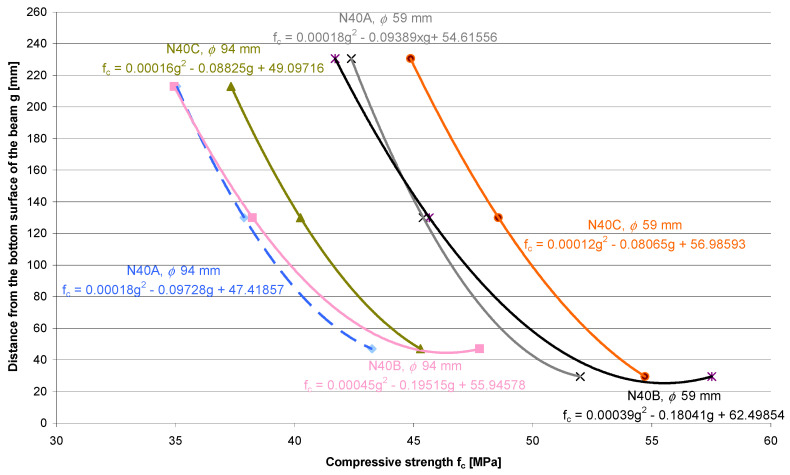
Distribution of the compressive strength of the 40 MPa concrete, which was determined across the height of the core samples with diameters of 59 mm and 94 mm.

**Figure 7 materials-13-05671-f007:**
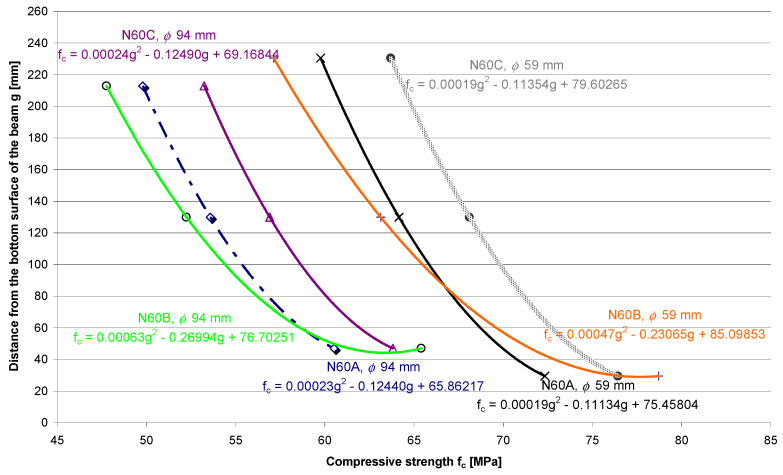
Distribution of the compressive strength of the 60 MPa concrete, which was determined across the height of the core samples with diameters of 59 mm and 94 mm.

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
