# Peer review of "Changes in the Compressive Strength of Concrete in Thin Horizontally Formed Slabs"

_materials, 2020, doi:10.3390/ma13245671_

Round 1

Reviewer 1 Report

The study on “Changes in the compressive strength of concrete in thin horizontally formed slabs” aims to verify the previous tests conducted by others, to establish the flaws from authors and/or supports many. The manuscript, indeed, established flaws but I think the authors supposed to show how to correct these flaws. If it has been shown in the manuscript, I think it’s not convincing, yet.

Authors should please check their first paragraph for language and reference appropriately, a classical example (line 29) must be supported with, at least, a reference.

Line 125 – 230,5 mm; comma or?

I think the manuscript should be checked for mistake/error and language. For instance, as you can see (line 151), authors could change this statement? We increase or increased? (line 153)

Author Response

The authors obviously agree with the comments of the Reviewer. The first paragraph of the introduction was corrected. In the summary, we added information on how to correct the defects shown. The entire text of the article was checked and corrected linguistically by a native speaker.

Line 125 (134 now) - We corrected the error and added an explanation.

Line 151, 153 (164, 166 now) - We changed these statements.

Reviewer 2 Report

The present paper studies the variation in concrete compressive strength tested using the destructive method across the thickness of a horizontally formed sample element. In this sense, the authors performed several samples with different diameters and manufactoring conditions.

In general lines the work is well structured but the redaction  can be revised in other to delete the first person of the text and other colloquial expressions, like "you".

Also the following changes are suggested to the authors:

  • Please include some references about the proccess of breaking proccess of the samples and include some photos about this proccess. Are there differences in the way the specimens break in each case?
  • The authors mention the verification of previous ultrasound tests, It would be interesting include the correlation between compression tests and ultrasound velocity. The authors mentione these tests but no information is provide.
  • Please revise the redaction of the chapter conclussion. It's no adequate mention explicitly another researcher.

Author Response

The authors obviously agree with the comments of the Reviewer. The entire text of the article was checked and corrected linguistically by a native speaker.

  • We did not take notes about the process of breaking process of the samples and we do not have any photos. We focused on finding the value of the compressive strength.
  • The problem between compression tests and ultrasound velocity was to study long ago and already been published. Therefore they did not write about it. It is quite a broad subject and it was not the subject of the manuscript.
  • The conclusion chapter was corrected and added a mention of a second author.

Reviewer 3 Report

Comments to the Authors

I have reviewed the manuscript “Changes in the compressive strength of concrete in thin horizontally formed slabs” by Jacek Szpetulski and Bohdan Stawiski. The work is very interesting, well-organised and needs some revisions to be published in Materials. Here are my comments.

  1. Revise some typos in the text.

  1. Add some comments in relation to the design applications.

  1. With reference to the regression analyses, please, add some coefficient to validate the effectiveness of the analysis (e.g., R2).

  1. It could be worth underlining that any experimental result is influenced by experimental uncertainty that can affect any engineering design/verification in addition to the aleatory and epistemic uncertainties, as discussed in the following reference. In addition, discuss how the results can impact the safety assessment of RC structures (as also commented in the following reference).

Castaldo P.Gino D.Mancini G. (2019) Safety formats for non-linear finite element analysis of reinforced concrete structures: discussion, comparison and proposals, Engineering Structures, 193, pp. 136-153.

Author Response

The authors obviously agree with the comments of the Reviewer.

  1. The entire text of the article was checked and corrected linguistically by a native speaker.
  2. The conclusion chapter was edited and added notes to regarding the design of concrete and the execution of concrete mix.

"In order to reduce the differences in the compressive strength across the thickness of a slab, it is necessary to ensure an appropriate consistency, and also a W/C ratio that will not liquefy the concrete mix. Thin structural elements made of concrete must be carefully formed by contractors who have experience in pouring and compacting concrete mixes."

  1. The coefficients R2 are equal to one, because the fitted trend lines for the three values of the average compressive strength are parabolas.

"In order to compare the compressive strength of the concrete at different levels, the values of the average compressive strength were calculated for five core samples (with a diameter of 59 mm and 94 mm) taken from each layer of the slab." - This exception added in Results and discussion chapter.

  1. In the Conclusions chapter added a mention how the results can impact the safety assessment of RC structures.

" If the fact that the compression zone is weaker by, e.g. 41.2 %, is taken into account, the load-bearing capacity of such a layer in the structure will decrease significantly. In monolithic ceiling slabs, the compression zone occurs in their upper parts, and the compressive strength of concrete is designed for this zone. In practice, this may mean that the concrete in the top zone does not meet the designer's assumptions."

Round 2

Reviewer 1 Report

This is having direction, I mean well written.

Author Response

The authors thank the Reviewer for a positive review.

Reviewer 2 Report

Although the content of the paper has improved, the authors should do a review of the writing of the article. There are expressions that should not appear in a scientific text. For that I encourage you to read other scientific articles.

Author Response

The authors obviously agree with the comment of the Reviewer. The authors did review of the writing of the article and deleted the expressions that should not appear in a scientific text. If the expressions are still, please indicate on which line.

Reviewer 3 Report

Comments to the Authors

I have reviewed the revised manuscript “Changes in the compressive strength of concrete in thin horizontally formed slabs” by Jacek Szpetulski and Bohdan Stawiski. The work is very interesting, well-organised and needs some revisions to be published in Materials. Here are my comments.

  1. Add some comments in relation to the design applications.

  1. It could be worth underlining that any experimental result is influenced by experimental uncertainty that can affect any engineering design/verification in addition to the aleatory and epistemic uncertainties, as discussed in the following reference. In addition, discuss how the results can impact the safety assessment of RC structures (as also commented in the following reference).

Castaldo P.Gino D.Mancini G. (2019) Safety formats for non-linear finite element analysis of reinforced concrete structures: discussion, comparison and proposals, Engineering Structures, 193, pp. 136-153.

Author Response

The authors obviously agree with the comment of the Reviewer.

  1. In the Conclusions chapter added the comment in relation to the design applications. - Line 232-236 and 241-244
  2. In the Conclusions chapter added a mention how the results can impact the safety assessment of RC structures and about uncertainty (as commented in the publication: Castaldo P., Gino D., Mancini G. (2019) Safety formats for non-linear finite element analysis of reinforced concrete structures: discussion, comparison and proposals, Engineering Structures, 193, pp. 136-153). - Line 224-232
